# Precise targeting for 3D cryo-correlative light and electron microscopy volume imaging of tissues using a FinderTOP

Marit de Beer [1,2,5], Deniz Daviran [1,2,5], Rona Roverts [1,2,5], Luco Rutten[1,2], Elena Macías-Sánchez[2,3], Juriaan R. Metz [4], Nico Sommerdijk [1,2✉] & Anat Akiva [1,2✉]

Cryo-correlative light and electron microscopy (cryoCLEM) is a powerful strategy to high resolution imaging in the unperturbed hydrated state. In this approach fluorescence microscopy aids localizing the area of interest, and cryogenic focused ion beam/scanning electron microscopy (cryoFIB/SEM) allows preparation of thin cryo-lamellae for cryoET. However, the current method cannot be accurately applied on bulky (3D) samples such as tissues and organoids. 3D cryo-correlative imaging of large volumes is needed to close the resolution gap between cryo-light microscopy and cryoET, placing sub-nanometer observations in a larger biological context. Currently technological hurdles render 3D cryoCLEM an unexplored approach. Here we demonstrate a cryoCLEM workflow for tissues, correlating cryo-Airyscan confocal microscopy with 3D cryoFIB/SEM volume imaging. Accurate correlation is achieved by imprinting a FinderTOP pattern in the sample surface during high pressure freezing, and allows precise targeting for cryoFIB/SEM volume imaging.

[1] Electron Microscopy Center, Radboud Technology Center Microscopy, Radboud University Medical Center, Nijmegen, The Netherlands. [2] Department of Medical Biosciences, Radboud University Medical Center, Nijmegen, The Netherlands. [3] Department of Stratigraphy and Paleontology, University of Granada, Granada, Spain. [4] Department of Animal Ecology and Physiology, Radboud Institute for Biological and Environmental Sciences, Faculty of Science, Radboud University, Nijmegen, The Netherlands. [5] These authors contributed equally: Marit de Beer, Deniz Daviran, Rona Roverts. ✉email: nico.sommerdijk@radboudumc.nl; anat.akiva@radboudumc.nl

Three dimensional (3D) high resolution imaging is crucial for understanding the structural organization and functioning of cells and tissues[1,2]. For 2D cell cultures, cryo-electron tomography (cryoET) has become available to resolve the structure of proteins within their native cellular environment[3,4]. In this context there is an increasing research focus on cryo-correlative light and electron microscopy (cryo-CLEM), where fluorescence microscopy aids in localizing the area of interest for high resolution imaging within the cell[5–10], and cryogenic focused ion beam/scanning electron microscopy (cryoFIB/SEM) is used to prepare thin lamellae for cryoEM or cryoET investigation[11].

Currently, this cryo-CLEM approach is almost exclusively used for the investigation of 2D cell cultures, that are plunge-frozen on a TEM grid. Here, the cell topography and added fiducial markers provide clearly recognizable features that allow straightforward correlation between light and electron images during FIB lamella preparation for cryoEM, where the recently introduced in-SEM-chamber light microscopy increases the success rate of the lamella preparation procedure[12]. However, 2D cell culture models do not suffice for addressing more complex questions that involve cell-cell or cell-matrix interaction, and 3D biological model systems are required[13–16]. The increasing demand for such 3D model systems, including organoids, also demands a different approach to imaging and sample handling.

Plunge freezing cannot be used for tissues due to the limited safe freezing depth (<5 μm), but samples up to 200 μm in thickness can be vitrified in their native state by high pressure freezing (HPF)[17,18]. However, in contrast to cryoCLEM on plunge frozen 2D cell cultures,[7,19] the featureless ice surface of HPF tissue samples prohibits the precise correlation of light and electron images prior to FIB/SEM volume imaging. This is exemplified in the recent demonstration of cryoCLEM for cryo-lift out and cryoET for nematode tissue, where the 2D correlation between light and electron images was limited by the absence of recognizable features[20].

Nevertheless, a holistic investigation of tissues requires the 3D correlation and analysis of volumes with dimensions that considerably exceed those which can be investigated with cryoET (<200 nm). This can be achieved through cryoFIB/SEM volume imaging, that allows the detailed nanoscale investigation of vitrified samples with multi-micron dimensions[21]. Nevertheless, to date volume imaging of tissues has mainly been performed using room temperature FIB/SEM[22,23] while only few examples exist of the use of 3D cryoFIB/SEM[14,21,24–29] and cryoCLEM volume imaging of tissues is still unchartered territory.

CryoFIB/SEM uses unstained, vitrified samples, while room temperature CLEM volume imaging of tissues so far uses heavy metal staining of plastic embedded samples[9]. Although this produces volume stacks that outclass the raw cryoFIB/SEM data in signal to noise ratios, the use of stained, dehydrated, embedded samples also has considerable drawbacks for CLEM. One is the fluorescence quenching by the staining agents, which necessitates the use of an additional imaging modality (e.g. X-ray computational tomography) for localization of the area of interest[30–32]. Another one is the need to use microtomy to prepare a starting plane for the FIB milling. A third one is the distortion of the volume due to anisotropic shrinkage of the embedding medium, which impacts the morphological fidelity of 3D images[9]. These drawbacks can be overcome by cryogenic fixation and imaging, allowing the visualization of cells and tissues in their native state in their natural environment[14,21,24–29].

Here we demonstrate a targeted cryoCLEM workflow for tissues, in which cryogenic confocal fluorescence imaging of millimeter scale volumes is correlated to 3D cryogenic electron imaging directed by a patterned surface generated during high pressure freezing (HPF). We apply this workflow for correlative imaging at the tissue level studying the mineralization process in scales of zebrafish as a model system for 3D organized organs. In these scales elasmoblasts[33], whose proteomic profile is near-analogous to human osteoblasts (bone forming cells), produce the extracellular matrix (ECM) that is only partially mineralized to maintain the balance between rigidity and flexibility for both protection and movement[33]. As the elasmoblasts remain active and vital for periods of hours after being removed from the skin, the scales form an interesting model to study bone formation processes[34,35].

By uncompromised imaging of tissues in their near-native state over all relevant length scales, from the millimeter down to the nanometer level, this workflow opens up future avenues to study structure-function relations of biological materials, in health and disease.

## Results

**General work flow**. As the zebrafish scales have millimeter dimensions with a thickness in the order of tens of micrometers, we use HPF for the vitrification. Cryo-Airyscan confocal microscopy (CACM)[19] is used to locate the region of interest (ROI) for 3D cryoFIB/SEM imaging as well as to determine the height, width and depth of the target volume in high pressure frozen tissue samples, however, registration of the two imaging modalities is complicated by the featureless surface of the vitrified ice layer[20]. By using a FinderTOP HPF carrier[36,37], a square grid pattern is imprinted in the ice surface during the vitrification, which is recognizable by both light and electron imaging modalities. Subsequent correlation is achieved computationally through rotational and translational alignment of the imaging modalities. As we are working with native tissue, without beads as fiducial markers, the fluorescent features of the cells are used as recognition points for the fine alignment of the images. 3D cryoFIB/SEM volume imaging is followed by image processing including removal of FIB/SEM artifacts (curtaining, charging), image alignment and contrast enhancement, to reveal the structural details of the sample in its near native state. Comparison of the two imaging modalities shows that, in contrast to room temperature CLEM, no deformation of the tissue occurs during the imaging procedure.

**Sample preparation**. Regenerating scales were harvested after 9 days from transgenic zebrafish which have nls (nuclear localization signal)-GFP expression under the control of SP7[38]. SP7 signals the production of osterix, a transcription factor that marks osteoblast formation. Additionally, the freshly harvested scales were fluorescently stained to label the Nucleus (Hoechst) and Mitochondria (MitotrackerRed). After staining, the scales were transferred to a HPF carrier (Ø 3 mm, depth 50 μm) which was topped up with a thin (~8 μm) layer of 10% dextran solution.

The carrier was then closed with a FinderTOP, a typical B type HPF carrier that on its flat side has a matrix with letters and numbers engraved. During vitrification, the engraved square 200 × 200 μm grid pattern of 4 μm high lines becomes imprinted in the sample surface (Fig. 1). The surface relief of the pattern is not only visible in the FIB/SEM, but also in reflection mode cryo-light microscopy due to surface scattering. This is essential for navigation of the otherwise featureless sample surface in cryoFIB/SEM[20]. Prior to the freezing, a sacrificial layer of the bilayer membrane forming lipid phosphatidylcholine is applied to the FinderTOP surface. During opening of the carrier, this directs the progressing fracture plane through the weakly interacting lipid bilayers, rather than through the vitrified sample, which would cause loss of the imprinted pattern.

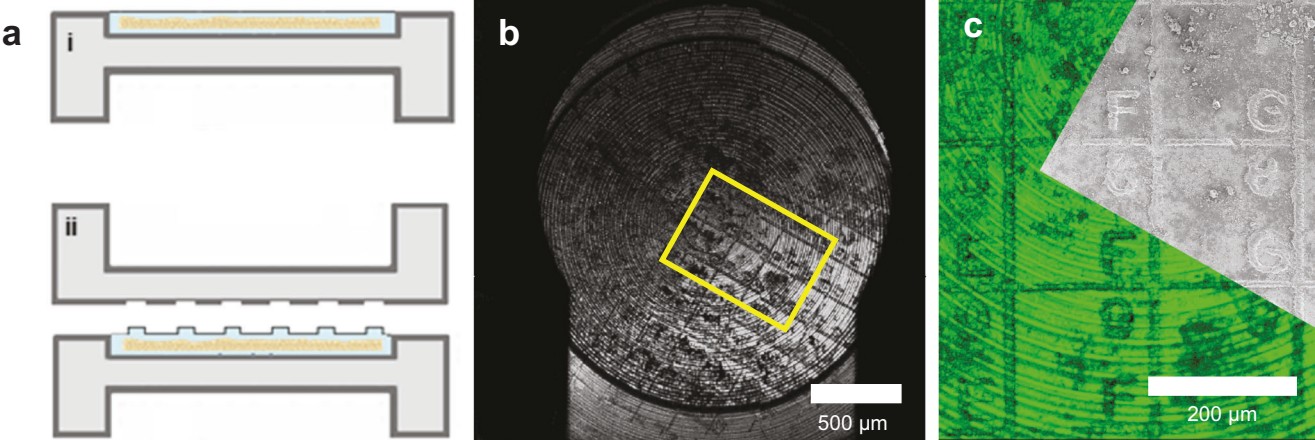

**Fig. 1 FinderTOP for aligning cryo-light and electron imaging modalities. a** Schematic representation of (i) a HPF carrier representing the featureless surface of tissue sample after vitrification using a conventional flat top carrier compared to (ii) vitrification using the FinderTOP, showing the imprinted pattern on the ice surface. **b** Reflection image of the vitrified sample using cryo-light showing the FinderTOP pattern. **c** Overlay of the reflection image in the confocal fluorescence microscope (green) with the SEM image in the FIB/SEM (gray). Letters and numbers can be seen in both modalities, like the highlighted "8", "9", "F" and "G".

**3D Cryo-fluorescence microcopy**. After vitrification, the sample in its 3 mm carrier was loaded into a universal cryo-holder that fixes its position in place, aiding the sample orientation between imaging modalities (Fig. 1). The universal holder was then placed in a cryogenic imaging adapter and positioned under an upright confocal laser scanning microscope with Airyscan superresolution[5]. To keep registration of the position of the ROI relative to the rest of the sample we built up a series of images with increasing resolution. First, a 2D low resolution overview image (pixel size 624 nm) was recorded of the entire sample for navigation, where the ice surface with FinderTOP pattern was visualized with reflection microscopy (Fig. 2a-i) and the scales with fluorescence microscopy (Fig. 2a-ii–v). This information was stored in an image correlation program to align and overlay the images and to facilitate further correlation with other imaging modalities.

As signal-to-noise ratios were low due to the reflection of the ice surface and the carrier bottom, an additional intermediate imaging step (pixel size x-y: 123 nm, z: 3000 nm) was used to locate the ROI with higher precision (Fig. 2b). Here, the ice surface showed an imprint a the FinderTOP letter "H" (Figs. 2b-i and 3a-ii) and although there is a high level of background, mitochondria, nuclei and SP7 positives cells are visible (Fig. 2b-iii–v). After overlaying the images in the correlation software, a 3D high resolution (pixel size x-y: 66, z: 360 nm) image of the ROI was recorded with CACM for navigation in 3D FIB/SEM volume imaging (Fig. 2c-i–iv). As the tissue sample is covered with a layer of dextran to fill up the volume of carrier above the scale, also high resolution 3D reflection images were recorded to determine the distance of the ROI with respect to the carrier sample surface (Fig. 2d).

**3D cryoFIB/SEM volume imaging**. After transfer to the FIB/SEM, overview secondary electron (SE) images at different magnifications were recorded to align the cryo-SEM view of the FinderTOP surface imprint with the optical overview reflection image of the ice surface and the high resolution CACM image of the ROI within the correlation software (Fig. 3a, b). For fine alignment, a FIB image taken at the coincidence point (with the SEM) was overlayed with already aligned SEM and optical images; (Fig. 3c). This enables stage registration and allows using the fluorescence micrographs to navigate to the ROI within the FIB/SEM.

The regions for milling and volume imaging were selected based on the high resolution cryo-fluorescence image and the

start of the volume stack is marked in the correlation software (Fig. 3d). For 3D cryoFIB/SEM volume imaging a trench for SEM observation was milled using a high FIB probe current, where the target depth (30 μm) was determined from the high resolution 3D refection images.

Here, we note that due to optical aberrations introduced by the air-ice interface, the z-distance between the sample surface and the upper surface of the scale observed in the reflection/fluorescence images (5.4 μm) deviated from the actual depth (8.9 μm) as observed in cryoFIB/SEM (Supplementary Fig. S1).

The resulting cross section was then polished with a low energy probe to create the first surface for serial surface imaging. Subsequently a stack of 97 images was created by alternatingly recording secondary electron (SE) images (pixel size 18 nm) and FIB milling (layer thickness 30 nm), generating a 3D image stack of 36 × 27 × 2.8 μm (Fig. 3e, f).

**Image processing**. Raw cryoFIB/SEM image stacks suffer from several features that prohibit their direct interpretation and hence their correlation with other imaging modalities (Supplementary Fig. S2). Firstly, the electron images are recorded from flat featureless surfaces where contrast relies on the variation in electron-sample interaction related to local differences in composition[21]. Next to the fact that these differences are small in biological materials, the low signal-to-noise ratio related to the low applied electron dose further limits the interpretability of the images. Secondly, the image acquisition process itself produces several artifacts, such as inhomogeneous FIB milling (vertical narrow striping: "curtaining"), local accumulation of electrons in non-conducting surfaces (horizontal broad striping: "charging") and the skewed brightness across the images due to detector imperfections (uneven illumination).

Using the correlation software package (for details, see materials and methods), first the curtaining was removed after which the local charge imbalance and uneven illumination were simultaneously corrected. This then allowed efficient alignment of the stack that was subsequently denoised after which a final local contrast enhancement step was applied. The processed FIB/SEM images (Fig. 4a, Movie 1 and 2)[39] now presented a clear cross-sectional view of the scale according to its previously described layered build up[33], revealing the layer of active elasmoblasts on top of layers of collagen with different orientation and structure and below that a layer of bone-like mineralized collagen

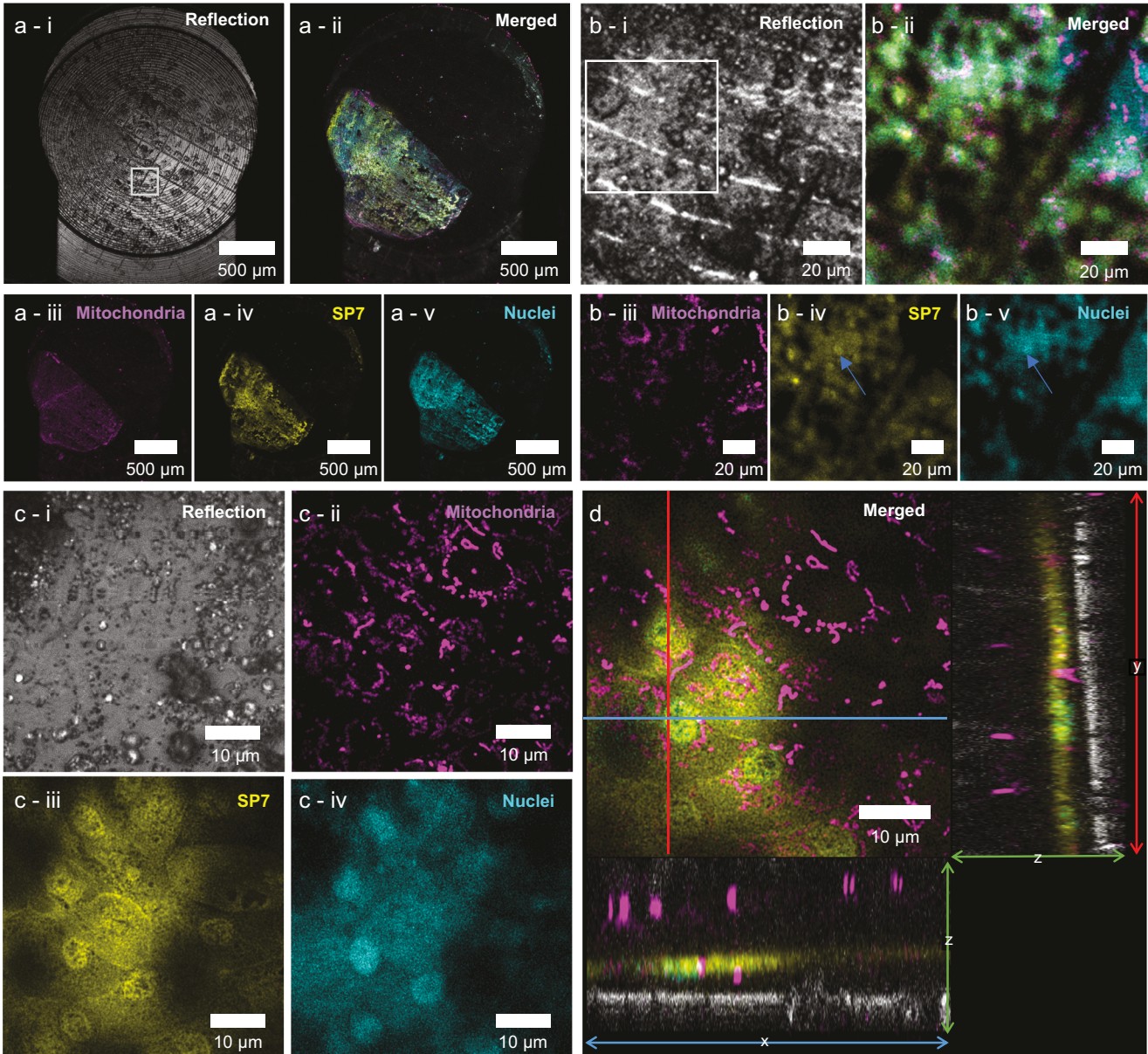

**Fig. 2 Cryo fluorescence microscopy of the scale. a** Overview images of the scale taken with the 5× objective to generate a complete overview of the carrier. **a**-i reflection mode shows the FinderTOP imprint. White box indicates the medium resolution area in **b**. **a**-ii Composite fluorescence image taken with the 5× objective, showing the scale for the different probes in the three channels. **a**-iii–v Images of the three different fluorescence channels. **b** Medium resolution imaging around the region of interest (ROI) using the 10× objective. **b**-i Reflection image; the white box denotes the ROI in **c**. **b**-ii Composite image showing the three fluorescent probes and the FinderTop imprint. **b**-iii–v Images of the individual fluorescence channels. **c**-i–iv High resolution images of the ROI taken with the 100× objective. Z- projection images (11 slices) of the reflection mode and each fluorescent channel. **d** Orthogonal views of the ROI indicated in **c**. The image shows the active osteoblast layer with the mitochondrial network. The ortho slices show the cross-sectional planes (yz and xz), where also the ice surface (reflection channel - gray) and the mitochondria in the posterior epithelial cell layer are observed. The ortho slices images are used to calculate the absolute depth of the ROI.

(Supplementary Fig. S3). The elasmoblast layer shows cells with well-preserved ultrastructure and with intact organelles (Fig. 4b-i–iv). 3D visualization afforded the details of the cell layers, including the arrangement of the different organelles with respect to the collagen – cell interface and the particulate structure of the mineralization front. (Movie 1 and 2)[39].

**Correlation between light and electron imaging**. Since the imaging resolution in cryo-fluorescence microscopy is the limiting factor, the alignment of the two imaging modalities is

performed in x-y (imaging) plane of the CACM. For the correlation of the imaging modalities the FIB/SEM data set was re-sliced, and individual corresponding 2D fluorescence - electron image pairs were corrected for the difference in the pixel size between the two imaging modalities. Mitochondria in the elasmoblast layer were selected as recognizable features in both modalities and registered visually following a rigid transformation approach (using only translation and rotation in ZEN Connect) (Fig. 5). Due to the difference in z-slice thickness between fluorescence (360 nm) and FIB/SEM (20 nm) several electron images in the stack over a range of 200 nm were matched with the

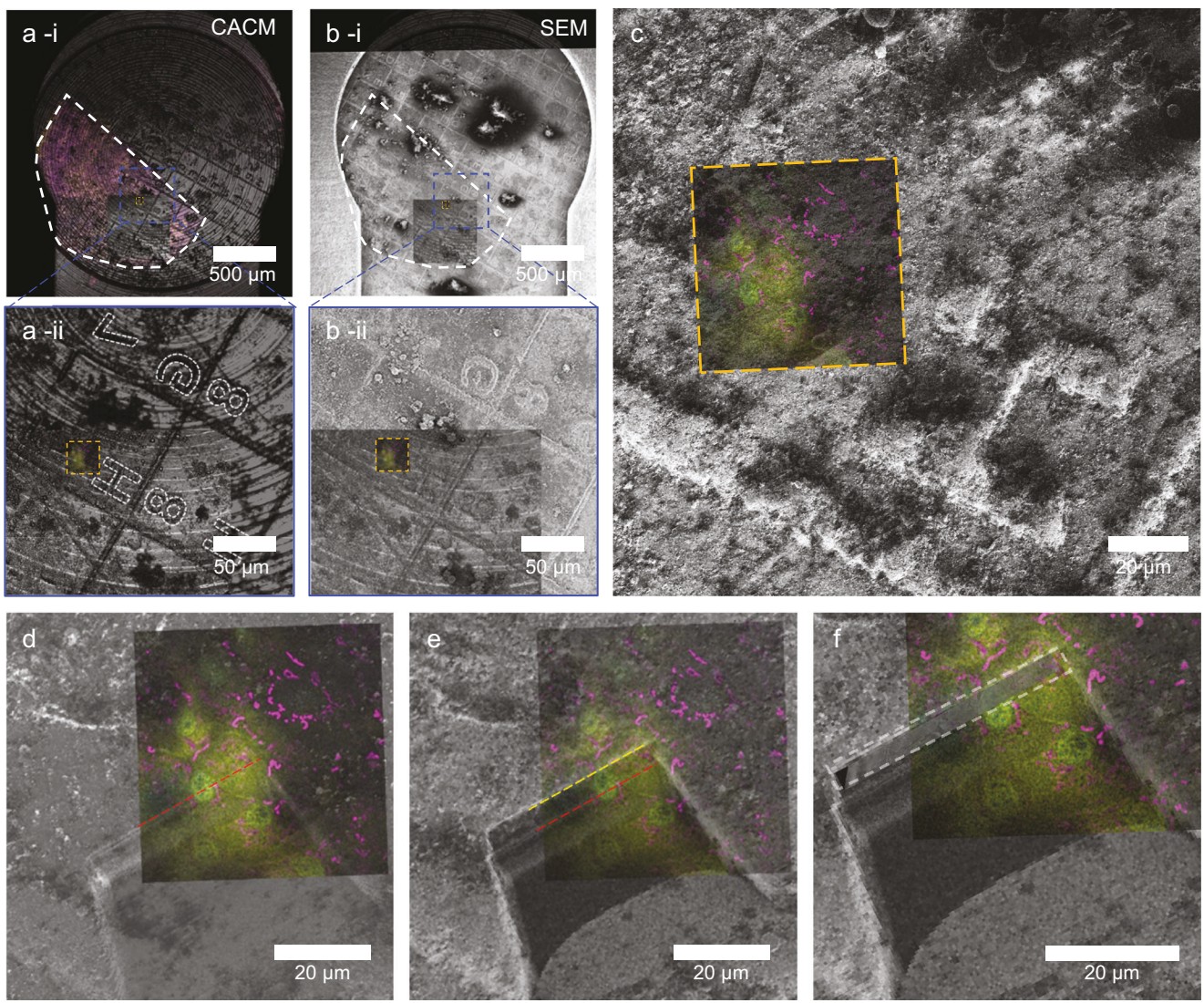

**Fig. 3 Localization of the region of interest (ROI) in cryoFIB/SEM. a** (i) Low magnification CACM image of the sample showing the FinderTOP imprint using reflection microscopy. The scale is highlighted as white outline. (ii) Overlay of medium resolution reflection microscopy with the ROI (orange box). This shows the ROI location in square "7H". **b** (i) Low magnification SEM image of the sample, showing the FinderTOP imprint. The scale is highlighted as white outline (ii) Alignment of the SEM image with the reflection image and the ROI in high resolution CACM (orange box) using the FinderTOP pattern. **c** FIB image overlayed with the ROI, taken in the coincident point and after tilting the sample to 54° (FIB beam perpendicular to the surface). **d** Overlay high resolution CACM image with the high magnification FIB image after generating the trench. The start position of the volume stack is highlighted (red line). **e** Overlay of high magnification FIB image with high resolution CACM image of the ROI. The end position of the volume stack is highlighted (yellow line). **f** Overlay of the resliced volume stack (white box) with the high magnification FIB image and CACM image of the ROI.

same fluorescence slice. (Fig. 5c–f). Registration of the images in both light and electron imaging also demonstrated the accuracy of the correlation during the localization of the ROI in the FIB/SEM (Fig. 3f).

## Discussion

Here we demonstrate 3D cryoCLEM volume imaging of 3D tissue samples in their near-native hydrated state, where we are able to select regions based on 3D fluorescence microscopy data to obtain detailed local 3D structural information on cellular and extracellular matrix components, and overlay this with the fluorescence data in 3D. We employ high pressure freezing and cryoFIB/SEM, which allow instantaneous and artifact free cryo-fixation of large 3D volumes, and high fidelity volume imaging, respectively.

Our approach contrasts the more common cryoCLEM strategy which employs plunge freezing, but due to the limited safe

freezing depth (<5 μm; beyond which crystalline ice is formed that is detrimental for cryoTEM) is restricted to the investigation of 2D cell cultures. Moreover, FIB milling is commonly used for on-grid lamella preparation for cryoEM, but so far there is only one account of 3D FIB/SEM volume imaging of these plunge frozen 2D cell monolayers on TEM grids[19]. Indeed, currently FIB/SEM volume imaging in CLEM has only been used on fixed embedded samples[22], where anisotropic shrinkage of the plastic embedding matrix leads to deformation of the sample between different imaging modalities[9]. Although cryoFIB/SEM volume imaging is associated with inherent low contrast and acquisition related imaging artifacts, we demonstrate that these limitations can be overcome by applying a designated computational toolbox to remove these artifacts and to enhance contrast.

The accuracy of the overlay in cryoCLEM imaging directly depends on the resolution of the optical imaging modality used. In confocal cryo-fluorescence microscopy, resolution is limited by

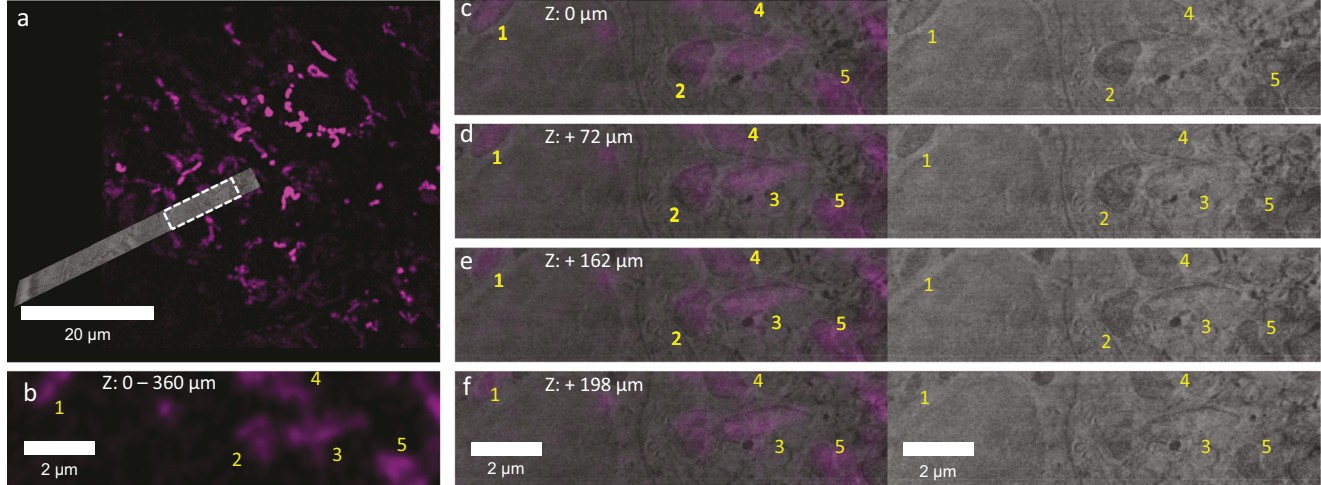

**Fig. 4 CryoFIB/SEM volume imaging. a** Single secondary electron (SE) image (inLens detector) showing the layered structure of the zebrafish scale. The cellular elasmoblast layer forms the top layer of the tissue and borders with the vitrified dextran solution. Below the cellular layer, there is a distinct collagen layer before reaching the bone-like mineral layer. **b**-i–iv Ultrastructural preservation in the elasmoblast showed by the selection of organelles (i; multivesicular body (MVB), ii; lysosome, iii; mitochondrion, iv; endoplasmic reticulum (ER)).

**Fig. 5 Correlation of cryoFIB/SEM and CACM images. a** Overlay images of single x-y slices from CACM and the resliced cryoFIB/SEM volume. The white dashed box shows the region in figures **b**–**f**. **b** Zoom-in of the fluorescent image showing five distinct regions labeled for mitochondria. Z-values indicate the total thickness of the single slice. **c**–**f** Image pairs at different depths of different resliced FIB/SEM images corresponding with the CACM image in **b** with and without the fluorescence overlay.

the numerical aperture of the objectives compatible with cryo-imaging[40], with an x-y resolution of 290 nm recently obtained using super resolution CACM of 2D cell cultures[5]. Also, as we show here, limitations in z-resolution of CACM (≈1 μm) also leads to an only modest accuracy in the z-direction. In addition, the optical aberrations introduced by refraction at the ice-water interface are different for different wavelengths, which further complicates the z-correlation of the different imaging modalities when using a multiple laser set-up (Supplementary Fig. S1). This asks for future technological improvement to achieve 3D targeting in larger volumes.

Whereas correlation with conventional super resolution approaches such as cryo-single molecule localization microscopy provides high resolution fluorescence information and high precision in correlation[8,41], it presents limitations with respect to the type and number of dyes that can be used. More precisely, it requires the use of specific photoactivatable dyes, is generally limited to two colors and requires intense laser powers[8,42]. Here we use CACM as it allows the parallel detection of multiple fluorescence signals, either endogenous labels or infiltrated dyes. In particular the possibility to use the infiltration of dyes is an attractive feature, as it opens up the approach to a wide range of wild type tissues, including human patient materials.

By the incorporation of cryoCACM, high pressure freezing and cryoFIB/SEM in a cryoCLEM workflow, we realized the multi-scale imaging of multicellular systems and tissues in their near-native state, from the millimeter down to the nanometer level. Accurate correlation is achieved by imprinting a FinderTOP pattern in the sample surface during high pressure freezing, and allows precise targeting of regions for cryoFIB/SEM volume imaging. Moreover, besides high pressure freezing, this cryo-workflow does not require sample preparation (other than infiltration of fluorescent dyes when wild type tissues are used) and provides correlated high quality 3D FIB/SEM images of cellular and extracellular details of multi-micron volumes. We expect this correlative cryogenic workflow to open the way to uncompromised targeted high resolution volume imaging of a wide variety of tissues.

## Methods

**Zebrafish and scale harvesting**. Zebrafish were maintained under normal husbandry conditions[43]. Transgenic line *Tg(Ola.sp7:NLS-GFP)* were anaesthetized using 0.1% (v/v) 2-phenoxyethanol and placed on a wet tissue containing system water and anesthetic. 9 days scales were plucked from 2-year-old females under a microscope with a watchmaker's tweezers from the midline of the lateral flanks near the dorsal fin.

**Fluorescent staining**. Fresh harvested scales (day 9 after regeneration) were fluorescently stained in fish culture medium for at least 2 h with the following dyes: Mitochondria (100 nM MitoTracker™ Red CMXRos, ThermoFisher, M7512) and Nucleus (1 μg/ml Hoechst 33342, ThermoFisher, H3570). Next, scales were then rinsed in fish culture medium.

**High pressure freezing**. Scales were immersed in 10% dextran (31349, Sigma) in demineralized water (MiliQ) and sandwiched between HPF carriers with 2 mm internal diameter[14]. Prior to creating the sandwich all carriers were intensively rinsed in pure ethanol and wiped with a cotton stick. The 0.05 mm cavity carrier (Art. 390, Wohlwend) was treated with a thin layer of 2% low-melt agarose, to serve as a spacer. A tailor-made grid labeled, flat-sided finderTOP (Alu-platelet labeled, 0.3 mm, Art.1644 Wohlwend) was treated with 1% L-α-phosphatidylcholine (PC, 61755, Sigma) in ethanol (1.00983.1000, Supelco). A monolayer of PC was applied by a pipetting a drop of 1% PC on the carrier, and subsequently remove all the liquid with the same pipet to have a thin layer remaining on the finderTOP. Note, when the L-α-phosphatidylcholine leads to white precipitation, clean the carrier more intensively and redo the treatment with phosphatidylcholine, because only a brown glow should be visible after treatment. The samples were then high pressure frozen using a HPM Live μ instrument[17] (CryoCapCell) and stored in liquid nitrogen.

The HPF carrier containing the vitrified scales was loaded into a universal cryo-holder (Art. 349559-8100-020, Zeiss cryo accessory kit) using the ZEISS Correlative Cryo Workflow solution, which fit into the PrepDek® (PP3010Z, Quorum

**Table 1 Cryo-light acquisition settings.**

| Target | Excitation | | | Emission |
| --- | --- | --- | --- | --- |
| | Laser (nm) | Laser power (%) | Master gain | Detection (nm) |
| SP7-GFP | 488 | 0.8 (14) | 850 (900) | 490–570 |
| Hoechst | 405 | 1.5 (25) | 900 (950) | 400–472 |
| Mitotracker | 561 | 1.0 (17) | 900 (950) | 571–620 |
| Reflection | 640 | 0.02 (0.2) | 600 | 538–700 |

Laser power and master gain was increased for the 100× objective (number between brackets).

technologies, Laughton, UK). This will improve the correlation between cryo-light and cryo-EM.

**Fluorescent cryo-imaging**. The universal cryo-holder containing the samples was transferred into the Linkam adapter to fit the LSM cryo-stage (CMS-196, Linkam scientific inc.), which was used with an upright confocal laser scanning microscopy (LSM 900, Zeiss microscopy GmbH) equipped with an Airyscan detector. First, overview images with low resolution (C Epiplan-Apochromat 5×/0.2 DIC, Zeiss; pixel size 624 nm) were made, to visualize the scales in fluorescence mode (settings described in Table 1) and the ice surface in reflection mode. Next, 3D images of the ROIs were recorded in medium resolution (C Epiplan-Apochromat 10×/0.4 DIC; pixel size x-y: 123 nm, z: 3000 nm). Finally, the ROIs were recorded in 3D with high resolution (C Epiplan-Neofluar 100×/0.75 DIC; pixel size x-y: 66, z: 360 nm) including the reflection microscopy images to measure the depth of the scales within the carrier. Settings had to be adjusted for this 100× objective. After the 100× images were recorded, a shift in Z was noted. The images could be corrected by adjusting the nuclear channel to the SP7 cell layer. Both channels have a nuclear staining, which makes correcting easy applicable (Supplementary Fig. S4). The shift adjustment was ~1.6 μm.

**CryoFIB/SEM**. The correlative cryo holder was mounted in the Quorum shuttle (Carl Zeiss Microscopy GmbH, Oberkochen, Germany) in the PrepDek® (PP3010Z, Quorum technologies, Laughton, UK). The shuttle was inserted into the Quorum cryo preparation stage with the transfer rod and sputter-coated with platinum, 5 mA current for 45 s, using the prep stage sputter coater (PP3010T, Quorum technologies, Laughton, UK). After coating, the shuttle was transferred into the Zeiss Crossbeam 550 FIB/SEM (Carl Zeiss Microscopy GmbH, Oberkochen, Germany) using the preparation chamber (PP3010T, Quorum technologies, Laughton, UK). Throughout imaging, the samples were kept at −160 °C and the system vacuum pressure was $2.1 \times 10^{-6}$ mbar.

After inserting the sample into the FIB/SEM chamber, overview SEM images (SE detector) were taken to align the SEM with the LSM reflection image of the ice surface within a single ZEN Connect project. This alignment enables the system registration which allows using the fluorescence signal to navigate to different regions of interest (ROI). To finalize the alignment of the ROI within the FIB/SEM, a FIB image (30 kV@10 pA probe under 54° tilt) of the surface was collected for a final alignment step with the SEM image.

Coarse trenches were milled for SEM observation using the 30 kV@30 nA FIB probe. If needed, cold deposition was performed with platinum for 45 s. The resulting cross section was polished using the 30 kV@3 nA probe. For volume cryoFIB/SEM imaging of the sample, a stack of 97 images was created by alternatingly recording electron images (both SE and BSE; pixel size 18 nm) and FIB milling (layer thickness 30 nm, FIB probe 30 kV@3 nA), generating a 3D image of $36 \times 27 \times 2.8$ μm. During imaging in-lens secondary electron imaging and back scatter electron images were simultaneously collected at low (2.30 kV) acceleration potential and low (50 pA) probe current to avoid beam damage. The EsB grid was set to 815 V. The image size was set to $2048 \times 1536$ pixels. For noise reduction line averaging with a line average count $N = 100$ at 100 ns pixel dwell time was used. The voxel size of the stack was $18 \times 18 \times 30$ nm. Images were saved in 16bit.tiff format.

**Image processing**. The cryoFIB/SEM images were processed using the EM processing toolbox of Zen Connect 3.5 (blue edition, Carl Zeiss) to correct for defects such as curtaining, misalignment and local charging, analogous to our previously reported approach[37], but with keeping the same processing settings for the whole stack. The same software was used for subsequent noise reduction and contrast enhancement. A summary of each processing step is as follows:

Curtaining: Removing the vertical stripes in the stacks was done using the "variational stationary noise removal" approach with only one filter in which the maximum level of noise allowed by the software was selected.

Charging: Elimination of the local charge imbalance and uneven illumination was achieved by using anisotropic gaussian background subtraction. Briefly, a mask with maximum sigma in the x direction and minimum sigma in the y direction was

created using the "Gauss" option in the toolbox. Subsequently, this mask was subtracted from the destriped image to obtain the charge balanced image. This approach simultaneously corrects any potential uneven illumination as well.

Alignment: To make sure that all the misalignments in the stack are in the same range, a manual alignment was initially done using the "Coarse Z-stack alignment" option in the toolbox. Subsequently, the smaller misalignments were corrected using "Z-stack alignment with ROI" option in the toolbox in which we only allowed for translational registration.

Noise Reduction: In order to improve the signal-to-noise ratio, noise reduction was performed using the "Denoise" option in the toolbox. The chosen denoising method was "Real wavelets" with a moderate strength.

Contrast enhancement: As the final processing step, the contrast was enhanced using "Enhance local contrast" option in the toolbox. To avoid over-amplification of homogeneous regions, we opted for a small clip limit.

**3D correlation**. The cryoFIB/SEM volume was resliced to be viewed in the plane corresponding to the CACM x-y plane. The two data sets (CACM with only the mitochondria's data and the resliced cryoFIB/SEM stack) were opened in Zen Connect 3.5 (blue edition, Carl Zeiss) for image overlay. The layered structure of the scale was instrumental in identifying the relevant cellular layer between the bone and the amorphous ice in both imaging modalities. As the two data sets shared the same known ROI through targeted imaging, only fine alignment, mainly in z direction, was required. The alignment was performed as follows: first, in the CACM stack 3-4 z slices in the mitochondrial layer were viewed, which showed minor differences in observed features due to the (relatively) poor z-resolution. The z slice with the strongest fluorescent signal was chosen for further manual alignment with the cryoFIB/SEM stack. During the alignment only translation and rotation operations were used.

**Reporting summary**. Further information on research design is available in the Nature Portfolio Reporting Summary linked to this article.

## Data availability
The datasets generated during and analyzed during the current study are available at: https://doi.org/10.5281/zenodo.7858909[39].

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

## Acknowledgements
The project was supported by an European Research Council (ERC) Advanced Investigator grant (H2020-ERC-2017-ADV-788982-COLMIN) to N.S. A.A. was also supported by a VENI grant from the Netherlands Scientific Organization NWO (VI.Veni.192.094). E.M.S. was supported by a Marie Skłodowska Curie Individual

Fellowship (H2020-MSCA-IF-2020- 101031624- DYNAMIN). The authors would like to acknowledge Endre Majorovits and Edwin Lamers from Carl Zeiss for technical support.

## Author contributions

N.S. and A.A. conceived, designed and supervised this study. N.S. and A.A. acquired funding and administrated the project. M.dB. performed the light microscopy and HPF experiments. D.D. performed the image processing and the correlation of light and EM data with support from M.dB. R.R. performed the cryoFIB/SEM with support from M.dB. L.R. and E.M.S. supported cryoEM experiments. J.M. provided the zebrafish scales. N.S. and A.A. wrote the manuscript with input from all authors.

## Competing interests

The authors declare no competing interests.

## Ethics approval

Animal research that is presented in this work has been conducted with ethical approval following the ARRIVE guidelines. We complied with all guidelines set in the European Directive 2010/63/EU of the protection of animals in research. Experiments were locally ethically reviewed by Radboud University (Nijmegen, NL) and performed under RU-DEC2014-059 and 2021-0013.
