## [Peer Review File · Communications Biology]

Reviewers' comments:

Reviewer #1 (Remarks to the Author):

In this paper, de Beer and colleagues show a workflow for cryoCLEM, which makes use of cryo-Airyscan confocal microscopy to target cryo volume imaging by FIB-SEM and lamella lift-out for cryo TEM. The targeting is aided by a finderTOP high pressure freezing carrier, which imprints a gridded pattern on the surface of the vitrified ice. This is used as a landmark at the FIB-SEM to retrieve the region(s) of interest identified by light microscopy.

The described workflow is aimed at facilitating one of the biggest bottlenecks in cryo electron microscopy analysis of large samples that require HPF for vitrification. The method shown is of high interest for the structural cell biology community. Although not entirely novel (FinderTOP shown in de Beer M et al (2021) Visualizing Biological Tissues: A Multiscale Workflow from Live Imaging to 3D Cryo-CLEM. *Microsc Microanal* 27:11–12. <https://doi.org/10.1017/S1431927621013635>; workflow used in Kepteijn et al (2022) *Nat Commun*. DOI: 10.1038/s41467-022-33054-w), the workflow was not previously described in details and it would deserve a new publication in my opinion. However, the current manuscript requires revision before publication.

The manuscript shows 2 different experiments, where cryo confocal is used for targeting i) cryo FIB-SEM volume imaging and ii) cryo lamella preparation and lift-out for cryo TEM.

The technical quality of the data shown about volume imaging is very high and the workflow sufficiently well described.

However, the lift-out and cryo TEM data unfortunately do not have the same quality. Moreover, they do not prove the success of the fluorescence-based targeting, which is the main topic of the study.

Finally, the cryo lift-out workflow would deserve a better description, as this is a technically demanding experiment and far from being routine.

Specific points to be addressed:

- Page 2. "fluorescence microscopy aids in localizing the area of interest for high resolution imaging within the cell5-7".

More references should be cited, perhaps more relevant than ref. 7.

- Page 2 "One is the fluorescence quenching by the staining agents, which necessitates the use of an additional imaging modality (e.g. X-ray computational tomography) for localization of the area of interest. Another one is the need to use microtomy to prepare a starting plane for the FIB milling. A third one is the distortion of the volume due to anisotropic shrinkage of the embedding medium, which impacts the morphological fidelity of 3D images28."

While these statements are in general correct, i) more reference should be added for the statements above and ii) ways to overcome these limitations have been reported, which could be mentioned (eg ref 20 and 28). Perhaps another more important point to make here to support the need for cryo FIB-SEM is the possibility of preserving the sample in a native state, without the use of heavy metals and resins and, in the case of lamellae and cryo ET, the possibility of in situ high resolution structural biology.

- Page 3. Sample preparation. "10% dextran solution".

Is it dissolved in the cell culture medium or in water? (same on page 13)

- How reproducible is the opening of the carrier sandwich to leave a visible grid? Is there any consideration regarding e.g. the sample or the cryoprotectant used?

- Figure 2.

In the current figure, it is difficult to read the intermediate resolution image shown in fig. 2b.

Especially it not clear what the SP7 and Hoechst nuclear staining are showing, as it looks like background to me. The authors should use a better image, highlight/label what the figure should show or remove it.

- The x,y,z axes in figure 2c should be shown as arrows.

- Figure 4 is very good, but should be complemented with a video showing the image stack.
- In the paragraph about correlation, please mention which software package/plugin was used for the transformation.
- Cryo lift-out and cryo TEM paragraph.

As I said above, this is the weakest part of the work. First of all, the workflow needs to be described in more details. Material and methods for this part only refer to a paper that does not show lift-out at all (ref 20). Moreover, the lamella generated is obviously too thick for a 2D TEM analysis and there is not much we can appreciate from fig. S3. My recommendation would be to use electron tomography here, in order to visualize cellular features and organelles. Last but not least, as the aim of the paper is "precise targeting", the authors need to prove that the lamella was generated in the desired location. An image showing correlation between the CACM and the cryo EM volumes is therefore necessary in my opinion.

- Material and methods. Cryo FIB-SEM. Page 14.

What does scan speed 1 correspond to? Please use dwell time instead.

Reviewer #2 (Remarks to the Author):

This manuscript describes a tailor-made "FinderTOP" carrier for HPF and a workflow of using this carrier for 3D correlative confocal (Airyscan) and volume FIB/SEM imaging of the zebrafish tissue. The "FinderTOP" imprints a grid pattern on the sample surface during HPF, and the pattern has sufficient signal to noise ratio in reflected light imaging and FIB/SEM imaging, thus can be used as the benchmark for the registration of these imaging modalities. In my opinion, this paper provides a new pathway for the correlative light and SEM imaging of tissues in 3D, and demonstrates the possibility of block-face volume using FIB-SEM. If the concerns below were addressed, I would suggest this manuscript for the publication in the journal.

Comments:

- (1) Fig.2 needs some modifications. The label for Fig.2d - ii is missing. Fig.2c and Fig.2d - i-iv are supposed to be the images of the same region, but they do not match (seem to be vertically flipped). The nuclei labelled in blue color (Fig.2d - iv) is not visible in the merged image (Fig.2c). The reflection image and the fluorescence image of individual color channels are all displayed in gray scale, which makes it difficult to read.
- (2) Fig.3 is in general not very clear. What region in Fig.3a corresponds to Fig.3b? Orange box and yellow box are really hard to be distinguished by color. From Fig.3b to Fig.3c, the shift of view and the slight change of magnification is misleading and confusing. If a higher magnification SEM image (Fig.3c yellow box) is required for the registration, I don't see the necessity of showing Fig.3b. In Fig.3e, the letter "H" is marked with dashed lines, whereas in other subfigures not. Why is that? All the reflected light, SEM and FIB images were in gray scale and displayed on top of each other, which makes it hard to follow.
- (3) Which software was used to register the light, FIB and SEM images upon the FinderTOP imprint? How the registration was done (manual alignment or automatic pattern recognition)?
- (4) Is the optical aberration uniform along the z-direction? Explicit experimental data was needed to prove that the optical aberration can be corrected by a linear scaling. Moreover, it is not clearly explained how the actual depth of 7.0 μm in FIB/SEM image was measured. Is the correction factor of 1.3 homogeneous across the whole sample?
- (5) Does the removal of curtaining and charging introduce artifacts?

(6) When correlating the light and SEM images in 3D, in the x-y direction, what is the precision of the "visual" registration upon recognizable features? In the z-direction, when choosing 1.5 μm deeper into the sample, how are the optical aberrations treated so that the light and SEM images were ensured to be the same plane?

Reviewer #3 (Remarks to the Author):

In this manuscript, Sommerdijk, Akiva et al report about a new tool to facilitate correlative light/electron microscopy (CLEM) in thick tissue samples that undergo high pressure freezing. The tool is FinderTOP, which is a square grid pattern, which imprints on the ice surface of the sample, during high-pressure freezing, a pattern recognizable by both light and electron imaging. The pattern allows aligning the images obtained by fluorescence and by scanning electron microscopy in samples that are undergoing cryo-FIB-SEM or lamella carving for the TEM. They exemplify the use of FinderTOP in the examination of a sample of zebrafish scale. Zebrafish scales are well-known and well-studied specimens, such that the results are not new in the sense of providing new information on the biological specimens.

The tool can undoubtedly be useful to an interested, although not very large community of users, who will welcome the possibility to purchase and use a FinderTOP during their correlative work.

I had, however, a hard time understanding what the authors wanted to show.

It is not straightforward from the beginning what a FinderTOP is, because the explanation comes only in the results. The reader understands then that FinderTOP is a brand name of an existing tool (Art.1644 Wohlwend (?)), but very few will recognize the item from the brand name. The idea behind using the FinderTOP, the novelty and the practice of the idea should be explained earlier, rather than just mentioning the name.

Figure 2b is supposed to show an active elasmoblast on top of the mineralizing matrix. I confess that I do not see anything in fig 2b. Furthermore, if figure 2d is a magnification of the white frame in figure 2b (?), figure 2d clearly shows many nuclei, not one cell, unless the cell is multinucleated. I am confused.

The description of the cryo-FIB-SEM imaging and image processing in the results is heavy and more appropriate for the methods section, because there are no novel aspects in the procedures. Besides, a bigger problem is in the quality of the CryoFIB/SEM imaging. The only reasonable part of the image is the cellular layer in Figure 4a. The other layers in the same figure do not contain any structural information, and the classification as collagen layer 1, 2, 3, 4 seems to be arbitrary. The cellular layers in Figure 4 b, c, d, are of a bad quality, such that they are not informative on the cellular structures and organelles as they should be. It is not clear why this should be so, as much better imaging can be achieved on cells.

I would recommend that the authors concentrate on the correlative procedure and on the new tool, which are valuable, without making the cryo-FIB-SEM imaging and reconstruction seem more important than they are, because this will make everyone notice that the results are not satisfactory.

Reviewer #1 (Remarks to the Author):

In this paper, de Beer and colleagues show a workflow for cryoCLEM, which makes use of cryo-Airyscan confocal microscopy to target cryo volume imaging by FIB-SEM and lamella lift-out for cryo TEM. The targeting is aided by a finderTOP high pressure freezing carrier, which imprints a gridded pattern on the surface of the vitrified ice. This is used as a landmark at the FIB-SEM to retrieve the region(s) of interest identified by light microscopy.

The described workflow is aimed at facilitating one of the biggest bottlenecks in cryo electron microscopy analysis of large samples that require HPF for vitrification.

The method shown is of high interest for the structural cell biology community. Although not entirely novel (FinderTOP shown in de Beer M et al (2021) Visualizing Biological Tissues: A Multiscale Workflow from Live Imaging to 3D Cryo-CLEM. *Microsc Microanal* 27:11–12.

<https://doi.org/10.1017/S1431927621013635>; workflow used in Kepteijn et al (2022) *Nat Commun*. DOI: 10.1038/s41467-022-33054-w), the workflow was not previously described in details and it would deserve a new publication in my opinion. However, the current manuscript requires revision before publication.

The manuscript shows 2 different experiments, where cryo confocal is used for targeting i) cryo FIB-SEM volume imaging and ii) cryo lamella preparation and lift-out for cryo TEM.

The technical quality of the data shown about volume imaging is very high and the workflow sufficiently well described.

However, the lift-out and cryo TEM data unfortunately do not have the same quality. Moreover, they do not prove the success of the fluorescence-based targeting, which is the main topic of the study.

Finally, the cryo lift-out workflow would deserve a better description, as this is a technically demanding experiment and far from being routine.

Specific points to be addressed:

- Page 2. “fluorescence microscopy aids in localizing the area of interest for high resolution imaging within the cell5-7”.

More references should be cited, perhaps more relevant than ref. 7.

Reference 7 is removed and replaced by the following papers: (Ref 5 – 10)

- 5 Sexton, D. L., Burgold, S., Schertel, A. & Tocheva, E. I. Super-resolution confocal cryo-CLEM with cryo-FIB milling for in situ imaging of *Deinococcus radiodurans*. *Curr Res Struct Biol* **4**, 1-9, doi:10.1016/j.crstbi.2021.12.001 (2022).
- 6 Gupta, T. K. *et al.* Structural basis for VIPP1 oligomerization and maintenance of thylakoid membrane integrity. *Cell* **184**, 3643-3659 e3623, doi:10.1016/j.cell.2021.05.011 (2021).
- 7 Klein, S., Wachsmuth-Melm, M., Winter, S. L., Kolovou, A. & Chlanda, P. Cryo-correlative light and electron microscopy workflow for cryo-focused ion beam milled adherent cells. *Methods Cell Biol* **162**, 273-302, doi:10.1016/bs.mcb.2020.12.009 (2021).
- 8 Tuijtel, M. W., Koster, A. J., Jakobs, S., Faas, F. G. A. & Sharp, T. H. Correlative cryo super-resolution light and electron microscopy on mammalian cells using fluorescent proteins. *Sci Rep* **9**, 1369, doi:10.1038/s41598-018-37728-8 (2019).
- 9 Hoffman, D. P. *et al.* Correlative three-dimensional super-resolution and block-face electron microscopy of whole vitreously frozen cells. *Science* **367**, doi:10.1126/science.aaz5357 (2020).
- 10 Tian, B., Xu, X., Xue, Y., Ji, W. & Xu, T. Cryogenic superresolution correlative light and electron microscopy on the frontier of subcellular imaging. *Biophys Rev* **13**, 1163-1171, doi:10.1007/s12551-021-00851-4 (2021).

- Page 2 “One is the fluorescence quenching by the staining agents, which necessitates the use of an additional imaging modality (e.g. X-ray computational tomography) for localization of the area of interest. Another one is the need to use microtomy to prepare a starting plane for the FIB milling. A third one is the distortion of the volume due to anisotropic shrinkage of the embedding medium, which impacts the morphological fidelity of 3D images²⁸.”

While these statements are in general correct, i) more reference should be added for the statements above and ii) ways to overcome these limitations have been reported, which could be mentioned (eg ref 20 and 28). Perhaps another more important point to make here to support the need for cryo FIB-SEM is the possibility of preserving the sample in a native state, without the use of heavy metals and resins and, in the case of lamellae and cryo ET, the possibility of in situ high resolution structural biology.

References have been added according to the referees suggestion.

- Page 3. Sample preparation. “10% dextran solution”.

Is it dissolved in the cell culture medium or in water? (same on page 13)

Dextran is dissolved in milliQ. We have added this to the method section on page 13 and added a reference.

- How reproducible is the opening of the carrier sandwich to leave a visible grid? Is there any consideration regarding e.g. the sample or the cryoprotectant used?

Getting the grid visible is a reproducible process. Samples should, like performing a normal freezing, fit in the carrier. The grid was visible in when 20% BSA, 17% Sucrose or 10% Dextran. Using cell culture medium was not tested.

The following sentence is added to the method section in the paper about the phosphatidylcholine (PC):

A monolayer of PC was applied by a pipetting a drop of 1% PC on the carrier, and subsequently remove all the liquid with the same pipet to have a thin layer remaining on the finderTOP. Note, when the L- α -phosphatidylcholine leads to white precipitation, clean the carrier more intensively and redo the treatment with phosphatidylcholine, because only a brown glow should be visible after treatment.

Figure: a small droplet was applied and then removed by using a pipet, leaving a very small amount of the phosphatidylcholine on the carrier. Left: finderTOP without intensive cleaning. Right: finderTOP cleaned with ethanol and a cotton stick.

- Figure 2.

In the current figure, it is difficult to read the intermediate resolution image shown in fig. 2b. Especially it not clear what the SP7 and Hoechst nuclear staining are showing, as it looks like background to me. The authors should use a better image, highlight/label what the figure should show or remove it.

At the intermediate resolution, there is a problem with fluorescence reflection caused by the ice surface in combination with the limiting Z-resolution. The images in figure 2 are improved by using single plane images and not Z-projections.

- The x,y,z axes in figure 2c should be shown as arrows.

We have add the arrows to show x,y,z in figure 2d (Previously fig 2c)

Figure 2 now looks as follows:

Figure 2. Cryo fluorescence microscopy of the scale. a) Overview images of the scale taken with the 5x objective to generate a complete overview of the carrier. a-i) reflection mode shows the FinderTOP imprint. White box indicates the medium resolution area in b). a-ii) Composite fluorescence image taken with the 5x objective, showing the scale for the different probes in the three channels. a-iii to a-v) Images of the three different fluorescence channels. b) Medium resolution imaging around the region of interest (ROI) using the 10x objective. b-i) Reflection image; the white box denotes the ROI in c). b-ii) Composite image showing the three fluorescent probes and the FinderTop imprint. b-iii to b-v) Images of the individual fluorescence channels. c-i to c-iv) High resolution images of the ROI taken with the 100x objective. Z- projection images (11 slices) of the reflection mode and each fluorescent channel. d) Orthogonal views of the ROI indicated in c). The image shows the active osteoblast layer with the mitochondrial network. The ortho slices show the cross-sectional planes (yz and xz), where also the ice surface (reflection channel - gray) and the mitochondria in the posterior epithelial cell layer are observed. The ortho slices images are used to calculate the absolute depth of the ROI.

- Figure 4 is very good, but should be complemented with a video showing the image stack.

Two movies are created with the data of figure 4 and captioned in the supporting information.

Movie 1. A video of figure 4 shows the zebrafish scale at the cryo-FIB/SEM imaging plane, imaged for 3,5 μm in depth. This movie shows the 3 elasmoblasts lying next to each other, underneath the cryo-protectant dextran solution. The cells are followed in depth by an elasmoidin layer (collagen) and at the bottom there is a mineral layer. During the movie, the main progression can be seen in the cellular organelles, like ER and mitochondria.

The images are recorded with a voxel size of $x=18\text{nm}$, $y=18\text{nm}$, $z=30\text{nm}$ and afterwards the serial sections were processed and aligned.

Movie 2. A video of figure 4, resliced to match the x-y view in CACM (see Fig. 5), starting at the top until a few micrometers into the elasmoidin layer. This shows the 3 elasmoblasts lying next to each other, underneath the cryo-protectant dextran solution. The cells are followed in depth by an elasmoidin layer (collagen). Here, we can see the different orientations of collagen.

- In the paragraph about correlation, please mention which software package/plugin was used for the transformation.

The correlation was done in zen Blue, using zen connect version 3.5 (Zeiss). This is now mentioned in the paper.

- Cryo lift-out and cryo TEM paragraph.

As I said above, this is the weakest part of the work. First of all, the workflow needs to be described in more details. Material and methods for this part only refer to a paper that does not show lift-out at all (ref 20). Moreover, the lamella generated is obviously too thick for a 2D TEM analysis and there is not much we can appreciate from fig. S3. My recommendation would be to use electron tomography here, in order to visualize cellular features and organelles. Last but not least, as the aim of the paper is “precise targeting”, the authors need to prove that the lamella was generated in the desired location. An image showing correlation between the CACM and the cryo EM volumes is therefore necessary in my opinion.

Indeed, the paper focusses on “*targeted, distortion free cryoCLEM workflow for tissues, in which 3D cryogenic fluorescence imaging of millimeter scale volumes is precisely correlated to 3D electron imaging directed by a patterned surface generated during high pressure freezing*” As currently we do not have the infrastructure, nor the possibility to perform high end cryoEM experiments, we added the lift-out results to the supporting information, just to show how our work could be applied more broadly in the field. In hindsight we agree with the referee that the cryo-TEM is not up to standard. As we will not be able to improve the cryoTEM images within the scope of this work, we had to decide to remove the part describing the lift-out and cryoTEM.

We adjusted the text accordingly.

- Material and methods. Cryo FIB-SEM. Page 14.

What does scan speed 1 correspond to? Please use dwell time instead.

The scan speed was change to Dwell time (100 ns) in the method section.

Reviewer #2 (Remarks to the Author):

This manuscript describes a tailor-made “FinderTOP” carrier for HPF and a workflow of using this carrier for 3D correlative confocal (Airyscan) and volume FIB/SEM imaging of the zebrafish tissue. The “FinderTOP” imprints a grid pattern on the sample surface during HPF, and the pattern has sufficient signal to noise ratio in reflected light imaging and FIB/SEM imaging, thus can be used as the benchmark for the registration of these imaging modalities. In my opinion, this paper provides a new pathway for the correlative light and SEM imaging of tissues in 3D, and demonstrates the possibility of block-face volume using FIB-SEM. If the concerns below were addressed, I would suggest this manuscript for the publication in the journal.

Comments:

(1) Fig.2 needs some modifications. The label for Fig.2d - ii is missing. Fig.2c and Fig.2d - i-iv are supposed to be the images of the same region, but they do not match (seem to be vertically flipped). The nuclei labelled in blue color (Fig.2d - iv) is not visible in the merged image (Fig.2c). The reflection image and the fluorescence image of individual color channels are all displayed in gray scale, which makes it difficult to read.

Fig 2 has been corrected and gray scaled images are replaced by color images (see also response to referee 1) .

(2) Fig.3 is in general not very clear. What region in Fig.3a corresponds to Fig.3b? Orange box and yellow box are really hard to be distinguished by color. From Fig.3b to Fig.3c, the shift of view and the slight change of magnification is misleading and confusing. If a higher magnification SEM image (Fig.3c yellow box) is required for the registration, I don't see the necessity of showing Fig.3b. In Fig.3e, the letter "H" is marked with dashed lines, whereas in other subfigures not. Why is that? All the reflected light, SEM and FIB images were in gray scale and displayed on top of each other, which makes it hard to follow.

Figure 3 is reorganized to tell the story more clearly. The figure now looks as follows:

Figure 3. Localization of the region of interest (ROI) in cryoFIB/SEM. a) (i) Low magnification CACM image of the sample showing the FinderTOP imprint using reflection microscopy. The scale is highlighted as white outline. (ii) Overlay of medium resolution reflection microscopy with the ROI (orange box). This shows the ROI location in square 7 H. b) (i) Low magnification SEM image of the sample, showing the FinderTOP imprint. The scale is highlighted as white outline (ii) Alignment of the SEM image with the reflection image and the ROI in high resolution CACM (orange box) using the FinderTOP pattern. c) FIB image overlaid with the ROI, taken in the coincident point and after tilting the sample to 54° (FIB beam perpendicular to the surface). d) Overlay high resolution CACM image with the high magnification FIB image after generating the trench. The start position of the volume stack is highlighted (red line). e) Overlay of high magnification FIB image with high resolution CACM image of the ROI. The end position of the volume stack is highlighted (yellow line). f) Overlay of the resliced volume stack (white box) with the high magnification FIB image and CACM image of the ROI.

(3) Which software was used to register the light, FIB and SEM images upon the FinderTOP imprint? How the registration was done (manual alignment or automatic pattern recognition)?

The 2D image alignment was done during image acquisition, using Zen Connect from Zeiss. Zen connect is based on manual alignment of images. Here, the finderTOP increases the accuracy of the alignment between the two modalities.

(4) Is the optical aberration uniform along the z-direction? Explicit experimental data was needed to prove that the optical aberration can be corrected by a linear scaling. Moreover, it is not clearly explained how the actual depth of 7.0 μm in FIB/SEM image was measured. Is the correction factor of 1.3 homogeneous across the whole sample?

We thank the referee for asking this important question. We realize now that the optical aberrations are wavelength dependent, so different for every laser channel. This means that without wavelength-specific corrections 3D overlay is not possible to have a 3D overlay of the two modalities.

We changed the text to read:

Here, we note that due to optical aberrations introduced by the air-ice interface, the z-distance between the sample surface and the upper surface of the scale observed in the reflection/fluorescence images (5.4 μm) deviated from the actual depth (8.9 μm) as observed in cryoFIB/SEM (Fig. S1).

We adapted figure 5 to now show only one channel (516 nm – mitochondria) and explain how the limited z-resolution in CACM leads to a match with cryo-FIB/SEM resliced topviews at different z-heights within the same CACM slice.

Figure 5. Correlation of cryoFIB/SEM and CACM images. a) Overlay images of single x-y slices from CACM and the resliced cryoFIB/SEM volume. The white dashed box shows the region in figures b-f. b) Zoom-in of the fluorescent image showing five distinct regions labeled for mitochondria. Z-values indicate the total thickness of the single slice. c-f) image pairs at different depths of different resliced FIB/SEM images corresponding with the CACM image in b) with (left) and without (right) the fluorescence overlay.

We changed the text to read:

Mitochondria in the elasmoblast layer were selected as recognizable features in both modalities and registered visually following a rigid transformation approach (using only translation and rotation in ZEN Connect) (Fig 5). Due to the difference in z-slice thickness between fluorescence (360 nm) and FIB/SEM (20 nm) several electron images in the stack over a range of 200 nm were matched with the same fluorescence slice. (Fig 5c-f). The absence of the need to apply anisotropic scaling or shearing to achieve the same registration with all image pairs, proves that no deformation (such as shrinkage) occurs during the cryo-workflow, in contrast to what is observed for room temperature CLEM of plastic embedded samples.⁹

We also added a supporting figure S1 showing how the optical aberrations are different in different channels, and that wavelength specific z-correction is needed to overlay in the x-z plane.

Figure S1. Overlay of cryoFIB/SEM and CACM in x-z view. (a-c) comparison of the x-z view in cryoFIB/SEM and CACM showing the optical aberrations introduced by refraction at the air-ice interface. (a) cryo-FIB/SEM slice. (b) x-z reflection image from resliced CACM stack corresponding to white dashed box in (a), showing the osteoblasts (yellow) and mitochondria (magenta). Arrow indicates the air-ice interface. Osteoblasts appear too close to the air-ice interface; mitochondria appear outside locations where cells are present. (c) x-y reflection image of the first z-position where the air-ice interface is detected, corresponding to the arrow in (b). (d-f) Overlay of cryoFIB/SEM and CACM data after wavelength specific z-correction. 640 nm reflection is used to indicate air-ice interface. To match with the cryoFIB/SEM the overlay of the 488 nm (osteoblasts) channel was stretched with 155% in the z-direction, the overlay of the 561nm (mitochondria) with 180%. (d) overlay with stretched 488nm image (e) with stretched 561nm image (f) with both 488nm and 561nm images.

We modified the discussion to say:

Also, as we show here, limitations in z-resolution of CACM ($\approx 1\mu\text{m}$) also leads to an only modest accuracy in the z-direction. In addition, the optical aberrations introduced by refraction at the ice-water interface are different for different wavelengths, which further complicates the z-correlation of the different imaging modalities when using a multiple laser set-up (Fig. S1). This asks for future technological improvement to achieve 3D targeting in larger volumes.

(5) Does the removal of curtaining and charging introduce artifacts?

As previously demonstrated, removal of the curtaining and charging can cause artifacts e.g. cross shape on lipid bodies, but these do not hamper the interpretation of the data.

- Vidavsky, N. *et al.* Cryo-FIB-SEM serial milling and block face imaging: Large volume structural analysis of biological tissues preserved close to their native state. *J Struct Biol* **196**, 487-495, doi:10.1016/j.jsb.2016.09.016 (2016)
- Spehner, D. *et al.* Cryo-FIB-SEM as a promising tool for localizing proteins in 3D. *J Struct Biol* **211**, 107528, <https://doi.org/10.1016/j.jsb.2020.107528> (2020)

Figure (from <https://doi.org/10.1016/j.jsb.2020.107528>). Typical cryo-FIB-SEM image processing steps. A) A typical raw image of a cryo-FIB-SEM stack showing lipid droplets, mitochondria, nuclear membrane and other cellular vesicles. B) Vertical curtaining stripes have been removed using a combined wavelet decomposition and Fourier filtering approach. C) Image in B denoised using 3D noise2void after drift correction. D) The denoised image in C after greyscale morphological reconstruction. Scale bar 500 nm.

(6) When correlating the light and SEM images in 3D, in the x-y direction, what is the precision of the “visual” registration upon recognizable features? In the z-direction, when choosing 1.5 μm deeper into the sample, how is the optical aberrations treated so that the light and SEM images were ensured to be the same plane?

Please see above at point (4)

Reviewer #3 (Remarks to the Author):

In this manuscript, Sommerdijk, Akiva et al report about a new tool to facilitate correlative light/electron microscopy (CLEM) in thick tissue samples that undergo high pressure freezing. The tool is FinderTOP, which is a square grid pattern, which imprints on the ice surface of the sample, during high-pressure freezing, a pattern recognizable by both light and electron imaging. The pattern allows aligning the images obtained by fluorescence and by scanning electron microscopy in samples that are undergoing cryo-FIB-SEM or lamella carving for the TEM. They exemplify the use of FinderTOP in the examination of a sample of zebrafish scale. Zebrafish scales are well-known and well-studied specimens, such that the results are not new in the sense of providing new information on the biological specimens.

The tool can undoubtedly be useful to an interested, although not very large community of users, who will welcome the possibility to purchase and use a FinderTOP during their correlative work. I had, however, a hard time understanding what the authors wanted to show.

It is not straightforward from the beginning what a FinderTOP is, because the explanation comes only in the results. The reader understands then that FinderTOP is a brand name of an existing tool (Art.1644 Wohlwend (?)), but very few will recognize the item from the brand name. The idea behind using the FinderTOP, the novelty and the practice of the idea should be explained earlier, rather than just mentioning the name.

We added a part in the introduction where we explain the FinderTOP better and added references.

By using a FinderTOP HPF carrier^{36,37}, a square grid pattern is imprinted in the ice surface during the vitrification, which is recognizable by both light and electron imaging modalities. Subsequent correlation is achieved computationally through rotational and translational alignment of the imaging modalities.

Figure 2b is supposed to show an active elasmoblast on top of the mineralizing matrix. I confess that I do not see anything in fig 2b. Furthermore, if figure 2d is a magnification of the white frame in figure 2b (?), figure 2d clearly shows many nuclei, not one cell, unless the cell is multinucleated. I am confused.

Figure 2 is modified to be more clear (see also response to referee 1 and 2)

The description of the cryo-FIB-SEM imaging and image processing in the results is heavy and more appropriate for the methods section, because there are no novel aspects in the procedures. Besides, a bigger problem is in the quality of the CryoFIB/SEM imaging. The only reasonable part of the image is the cellular layer in Figure 4a. The other layers in the same figure do not contain any structural information, and the classification as collagen layer 1, 2, 3, 4 seems to be arbitrary. The cellular layers in Figure 4 b, c, d, are of a bad quality, such that they are not informative on the cellular structures and organelles as they should be. It is not clear why this should be so, as much better imaging can be achieved on cells. I would recommend that the authors concentrate on the correlative procedure and on the new tool, which are valuable, without making the cryo-FIB-SEM imaging and reconstruction seem more important than they are, because this will make everyone notice that the results are not satisfactory.

We have removed some of the numbers from the Results to the Methods section. And added a freeze fracture SEM experiment to the Supporting Information showing the different collagen layers in the elastomoid, to support our interpretation of the cryo-FIB/SEM images. We have improved Figure 4 to show the cell layer in more details and to demonstrate the high quality of the freezing and imaging. Figure 4 now looks as follows:

Figure 4. CryoFIB/SEM volume imaging. a) Single secondary electron (SE) image (inLens detector) showing the layered structure of the zebrafish scale. The cellular elasmoblast layer forms the top layer of the tissue and borders with the vitrified dextran solution. Below the cellular layer, there is a distinct collagen layer before reaching the bone-like mineral layer. b - i to b - iv) Ultrastructural preservation in the elasmoblast showed by the selection of organelles (i; multivesicular body, ii; lysosome, iii; mitochondrion, iv; endoplasmic reticulum).

REVIEWERS' COMMENTS:

Reviewer #1 (Remarks to the Author):

The authors have addressed all the points I raised. I think the new figures have improved the presentation of the work and I am glad the authors agreed to remove the part about cryo-TEM, which did not have a sufficient quality and was not necessarily within the scope of the work.

I only have one further concern with the new version of the manuscript:

Figure S1 shows that the authors need to stretch the CACM data in the Z direction in order to match the cryoFIB-SEM data. I would therefore be careful with the statement "the absence of the need to apply anisotropic scaling or shearing to achieve the same registration with all image pairs, proves that no deformation (such as shrinkage) occurs during the cryo-workflow" (page 8). While I agree that this is not due to deformation of the sample, the authors in fact do apply anisotropic scaling (x,y scaling is different from z). This conclusion should be rephrased before publication.

Reviewer #2 (Remarks to the Author):

I thank the authors for their response to reviews, and I believe they have satisfactorily addressed the suggestions and criticisms.

Reviewer #3 (Remarks to the Author):

The quality of the data in the revised manuscript is much improved, and the explanations are more focused, which makes the whole manuscript clearer. In this sense, the manuscript is now acceptable.

We have changed the manuscript in accordance with the comments of the referee

We removed “distortion free” on page 2

We removed *“The absence of the need to apply anisotropic scaling or shearing to achieve the same registration with all image pairs, proves that no deformation (such as shrinkage) occurs during the cryo-workflow, in contrast to what is observed for room temperature CLEM of plastic embedded samples.”* on page 8.